# Understanding and Harnessing Sparsity for Unified Multimodal Models

## Abstract

Large multimodal models have achieved remarkable progress in both understanding and generation. Recent unified multimodal models integrate heterogeneous components so that a single framework can support both capabilities. This unification, however, can introduce inference inefficiency: a given task or sample may require only a subset of the model's full capacity. How this inefficiency differs across the understanding and generation components remains underexplored. In this work, we analyze unified multimodal model components with training-free pruning probes, covering both depth pruning and width reduction. We find that the understanding component is compressible for both understanding and generation tasks, with substantially greater tolerance in generation. In contrast, generation components are highly sensitive to static compression, with performance degrading sharply even at moderate compression ratios. Motivated by the dynamic activation patterns observed across samples, we propose Mixture-of-Experts (MoE) Adaptation for the generation component. The method partitions MLP neurons into experts and sparsely activates them to recover generation quality under a reduced active parameter budget. We first validate sparse activation through expert-frozen tuning and then show that fully trainable adaptation provides additional gains. On BAGEL and GenEval, the adapted model achieves performance comparable to the dense baseline while activating about half of the generation-component neurons; we further report latency and memory measurements to characterize realized efficiency rather than relying on activated-parameter counts alone. The code will be released upon acceptance.

## 1 Introduction

Large-scale multimodal models have recently achieved strong results in both multimodal understanding (Liu et al., 2023; Li et al., 2023; Dai et al., 2023; Lu et al., 2024; 2023) and generation (Ramesh et al., 2021; Saharia et al., 2022a; Peebles & Xie, 2023). Traditionally, these two tasks were studied through separate model families: *understanding models* for vision-language reasoning with textual outputs, and *generative models* for image synthesis. While effective for task-specific purposes, this separation limits the ability of a single model to reason over multimodal inputs and generate multimodal outputs within the same system. Recent research has therefore explored unified multimodal models (Wu et al., 2024; Chen et al., 2025; Deng et al., 2025; Liang et al., 2025; AI et al., 2025), which integrate heterogeneous components such as vision encoders (Dosovitskiy et al., 2021), language backbones (Dubey et al., 2024; Yang et al., 2024), and image decoders (Peebles & Xie, 2023; AI et al., 2025). These architectures support both understanding and generation in a single framework, making them a promising direction for general-purpose multimodal intelligence.

However, this unification comes at a substantial ***efficiency cost***, because many tasks or input samples may require only a subset of the unified model's capacity Frankle & Carbin (2018); Xu et al. (2021). Three observations suggest that this inefficiency should be studied at the component, task, and input levels: (1) ***Component-wise redundancy***: the understanding and generation components follow distinct computation patterns and serve different functional roles, leading to different levels of redundancy. (2) ***Family-specific activation***: inputs from different task families tend to activate only a limited subset of parameters during inference, leaving much of the shared capacity underutilized Kudugunta et al. (2021); Sarkar et al. (2023).

(3) **Input variability**: even within the same task, different input queries can activate different portions of the model, further complicating efficient compute allocation Liu et al. (2024); Luo et al. (2025).

Motivated by these factors, we systematically analyze how unified multimodal models allocate and use parameters across components, tasks, and inputs. We first adopt **training-free pruning** as a probing methodology (Gromov et al., 2025; Men et al., 2024; He et al., 2024; Ma et al., 2023; Xia et al., 2023), which lets us infer structural importance by removing structures and measuring the resulting performance change. We begin by analyzing the understanding components, which serve as shared modules that process inputs from multiple modalities and produce language representations or embeddings that guide the generation components. Our results demonstrate that the understanding components are highly compressible in multimodal generation tasks but less so in understanding tasks. We also observe clear family-specific activation patterns: understanding and generation inputs predominantly activate different model partitions, underscoring the need for family-aware activation.

In contrast, generation components (e.g., image generators) are far less tolerant to static compression. Their activation patterns vary substantially across samples and timesteps, so static pruning cannot accommodate the required activation shifts and leads to a sharp drop in generated image quality. To address this issue, we propose **Mixture-of-Experts (MoE) Adaptation**, where MLP neurons are partitioned into experts so that the model can selectively activate different neuron subsets for different inputs. To align the model with sparsely routed inference, we first introduce expert-frozen tuning, which freezes all experts and updates the router and other remaining parameters. Building on this initialization, we further perform end-to-end MoE training, yielding additional improvements. We validate this full training-aware adaptation as a BAGEL/GenEval case study: the adapted BAGEL model (Deng et al., 2025) matches the performance of the dense baseline while activating about half of the generation-component neurons. We additionally report measured latency and memory to distinguish reduced activated parameters from realized system speedup. Our contributions are threefold:

- We show that the understanding component in unified multimodal models has substantial structural redundancy and can be compressed more aggressively when it supports generation tasks.
- We introduce a training-free neuron partition analysis that uncovers family-specific activation patterns and redundancy within the understanding component, providing a basis for family-aware compression.
- To address the high compression sensitivity of the generation component, we design MoE Adaptation and validate it on BAGEL, preserving generation quality under sparse activation while reporting runtime and memory behavior.

## 2 Related Work

**Multimodal Models for Understanding and Generation**  Early multimodal research typically treated understanding and generation as separate tasks, leading to distinct architectural paradigms (Li et al., 2023; Dai et al., 2023; Zhu et al., 2023). On the one hand, multimodal large language models (MLLMs) extend language models to handle input tokens from multiple modalities, such as LLaVA (Liu et al., 2023), which augments the LLaMA backbone (Touvron et al., 2023a) with visual tokens for vision–language understanding. On the other hand, multimodal generative models use dedicated generators to synthesize high-fidelity visual outputs (Ramesh et al., 2022; Saharia et al., 2022b; Chang et al., 2023). For instance, Diffusion Transformers (DiT) (Peebles & Xie, 2023) show that iterative denoising with transformer-based backbones can generate high-quality images guided by natural language. More recently, several works aim to unify understanding and generation within a single framework (Wu et al., 2024; Chen et al., 2025; Zhang et al., 2025). BAGEL (Deng et al., 2025) employs a mixture-of-transformers design (Liang et al., 2025) to separate understanding and generation modules, while Ming-Omni (AI et al., 2025) uses a mixture-of-experts backbone with multimodal understanding and modality-specific decoders for generation. Although these unified models are versatile, their architectural complexity creates new efficiency challenges that remain underexplored.

**Model Compression toward Parameter Efficiency**  As large language models continue to scale, their size introduces substantial redundancy and raises practical challenges for deployment. Network pruning (Liu et al., 2019; Cheng et al., 2024) identifies and removes redundant structures, improving parameter efficiency

and reducing inference cost. For instance, (Gromov et al., 2025) demonstrated that many deep layers in large language models are relatively unimportant and that comparable performance can still be maintained after removing them. Within individual layers, *width compression* provides a complementary way to reduce parameter count by shrinking intermediate or hidden dimensions (Ma et al., 2023; Xia et al., 2023). While unimodal compression techniques can be transferred to vision-language models that take multimodal inputs and produce language responses (Sung et al., 2024; Lin et al., 2024; He et al., 2025b), it remains unclear how such methods behave in unified models that contain both understanding and generation components. Building on prior efforts, we study redundancy in unified multimodal models where heterogeneous components play distinct roles. This component-aware perspective motivates compression and sparse activation strategies aligned with both multimodal understanding and generation.

## 3 Preliminaries: Unified Multimodal Models

Unified multimodal models integrate multimodal understanding and generation within a single framework.

**Understanding** Given a unified multimodal model, let $\mathbf{x}$ denote the multimodal input, and $\mathbf{y}$ denote the corresponding output. For understanding tasks, the model predicts textual outputs in an autoregressive manner:

$$p(\mathbf{y}_{\text{und}} \mid \mathbf{x}; \theta_{\text{und}}) = \prod_{t=1}^{T} p(y_t \mid y_{<t}, \mathbf{x}; \theta_{\text{und}}), \tag{1}$$

where $\theta_{\text{und}}$ denotes the parameters of the understanding component and $\mathbf{y}_{\text{und}} = (y_1, y_2, \ldots, y_T)$ denotes the output token sequence.

**Generation** For generation tasks, the unified model uses the understanding component to process an instructional input $\mathbf{x}_{\text{inst}}$ (e.g., text prompts and reference images), producing conditional features $f_{\text{und}}(\mathbf{x}_{\text{inst}}; \theta_{\text{und}})$. The generation component, parameterized by $\theta_{\text{gen}}$, synthesizes the output $\mathbf{y}_{\text{gen}}$ conditioned on these features and an additional generative input $\mathbf{z}$ (e.g., random noise in diffusion models):

$$\mathbf{y}_{\text{gen}} \sim p(\mathbf{y}_{\text{gen}} \mid f_{\text{und}}(\mathbf{x}_{\text{inst}}; \theta_{\text{und}}), \mathbf{z}; \theta_{\text{gen}}), \tag{2}$$

In unified multimodal models, the understanding component processes inputs from multiple modalities, while the generation component operates on non-text outputs such as image or audio synthesis. Because these components have different objectives and computational pathways, they can respond differently to compression.

## 4 Methodology

We first introduce training-free pruning techniques for probing architectural redundancy, then propose a training-aware MoE adaptation that uses dynamic sparsity to preserve performance under sparse activation. Figure 1 summarizes the two strategies.

### 4.1 Training-free Compression Strategies

**Depth Pruning via Layer Dropping** Transformer-based large language models consist of multiple layers, and increasing depth is an effective way to improve performance. However, depth can also introduce structural redundancy. Following (Gromov et al., 2025; Men et al., 2024; He et al., 2024), we measure the redundancy of a layer as:

$$S_l = \text{CosineSim}(\mathbf{x}_l, \mathbf{y}_l), \tag{3}$$

where $\mathbf{x}_l$ and $\mathbf{y}_l$ are the input and output of the $l$-th layer. Higher similarity indicates that the layer applies a smaller transformation and is therefore more redundant. This metric has been effective in unimodal LLMs such as Mistral (Jiang et al., 2023) and LLaMA (Touvron et al., 2023b; Dubey et al., 2024); here we use it to probe unified multimodal models.

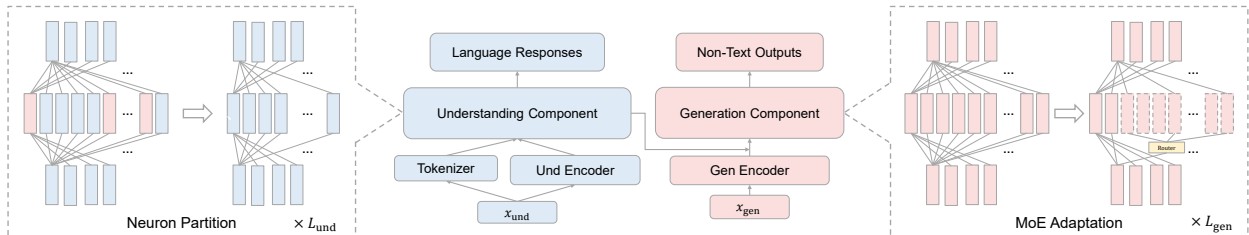

**Figure 1: Overview of our training-free compression and training-aware MoE adaptation strategies.**

**Width Reduction via Neuron Partition**    In addition to depth, width, particularly in MLP layers, is a major source of model capacity. In Transformer MLP layers, the input is expanded from dimension $d$ to $dm$, activating $dm$ hidden neurons before being projected back to $d$. This expansion increases capacity but can also introduce neuron-level redundancy. We therefore propose neuron partitioning, which separates hidden neurons into important and less important subsets, preserves the former, and prunes the latter for structured width reduction.

To measure neuron importance, we take inspiration from Wanda (Sun et al., 2024), which combines weight magnitudes and activation statistics, and extend it from unstructured weight-level pruning to a structured neuron-level metric. Given an input $x \in \mathbb{R}^{s \times d}$, the hidden activations $h \in \mathbb{R}^{s \times dm}$ and output $y \in \mathbb{R}^{s \times d}$ of a Gate-Up-Down MLP are:

$$h = \left(\mathrm{SiLU}(xW_g^\top)\right) \odot (xW_u^\top), \qquad y = hW_d^\top, \tag{4}$$

where $W_g, W_u \in \mathbb{R}^{dm \times d}$ are the gate-projection matrix and up-projection matrix, $h \in \mathbb{R}^{s \times dm}$ is the gated activation, and $W_d \in \mathbb{R}^{d \times dm}$ is the down-projection matrix. The contribution of the $i$-th neuron to the final output is:

$$\Delta y_i = h_i W_{d,i}^\top, \tag{5}$$

with $W_{d,i}$ being the $i$-th column vector of $W_d$. If the $i$-th neuron is pruned, the induced output error norm can be approximated by:

$$\|\Delta y\|_2 \approx \|h_i W_{d,i}^\top\|_2. \tag{6}$$

Given all inputs from the calibration dataset $\mathcal{D}$, the expected accumulated error of each neuron is used as its importance metric:

$$s_i = \mathbb{E}_{x \sim \mathcal{D}}\left[|h_i| \cdot \|W_{d,i}^\top\|_2\right], \tag{7}$$

where $|h_i|$ measures the average activation magnitude of the $i$-th neuron and $\|W_{d,i}\|_2$ quantifies its amplification effect on the output. Neurons with larger scores are retained, while those with smaller scores are removed. Unlike unstructured pruning, which zeros individual weights, our approach removes entire neurons. Concretely, this removes column $i$ from $W_d$ and row $i$ from both $W_u$ and $W_g$, yielding a structured reduction.

Unified models integrate diverse task families within a single architecture, and different families can activate different neuron subsets. Figure 2 illustrates the distinct neuron partitions within the understanding component. We first identify the top 50% of neurons by importance score for understanding and generation, respectively. We then measure the overlap subset between the two top-50% sets and report this overlap share relative to all neurons in the layer. Since the two top-50% sets have the same size, the one-side non-overlap share is simply 50% minus the

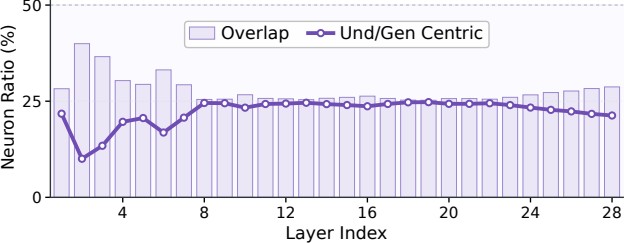

**Figure 2: Top-50% neuron-set overlap between understanding and generation calibration.**

overlap share. The relatively low overlap share suggests that understanding and generation inputs activate distinct neuron subsets within the understanding component. We therefore align calibration samples with the target task family before applying the neuron-level importance metric, which improves the identification of family-relevant neurons.

## 4.2 Training-Aware MoE Adaptation

Figure 3 visualizes active neurons (those consistently ranked within the top 50% by activation score) and inactive neurons (those that never enter the top 50%) across layers of the generation component over multiple prompts and time steps. Only a small subset of neurons remains consistently active throughout inference, while most neurons exhibit sample-dependent activation patterns. This input-dependent activation aligns with the motivation for Mixture-of-Experts (MoE) architectures. We leverage it through two stages: *Expert Partition* and *MoE Adaptation*.

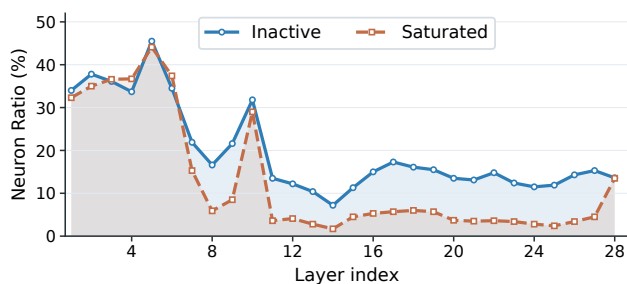

**Figure 3: Inactive and saturated neurons in the generation component across layers.**

**Expert Partition**  To separate universal and sample-specific capacity (He et al., 2023; Dai et al., 2024), we partition MLP neurons into shared and routed experts using cumulative importance over calibration samples. For each neuron, we compute its cumulative importance score using Equation 7. Neurons with scores above a threshold are selected as shared experts $E_s$, preserving features that consistently benefit multiple samples. The remaining neurons, whose relative importance is more sample-dependent, are allocated to routed experts $E_r^{(1)}, \ldots, E_r^{(n)}$ by ranked importance. Specifically, neurons are sorted in descending importance order and assigned to experts in alternating forward and reverse order (i.e., $E_r^{(1)}$ to $E_r^{(n)}$, then $E_r^{(n)}$ back to $E_r^{(1)}$), balancing total importance across experts.

**MoE Adaptation**  After expert partition, we insert one router per layer to dynamically select routed experts for each input. The MoE layer output is:

$$\text{MoE}(x) = f_{\mathcal{S}}(x) + \sum_{j \in \text{Top-}k(\mathcal{G})} \mathcal{G}_j \cdot f_{\mathcal{R}_j}(x), \tag{8}$$

where $\mathcal{G}$ denotes the gating function, and $f_{\mathcal{S}}$ and $f_{\mathcal{R}}$ are the transformations of the shared and routed experts. The original MLP layer is a special case of Equation 8, where all experts are selected and their gating scores are uniformly set to 1. To smooth the transition from dense to sparse activation, we relax the constraint that gating scores sum to 1 and reparameterize them as $(1 + \text{Router}(x))$, with the router initialized to zero. The conversion partitions the original MLP weights into experts rather than duplicating a full dense MLP for every expert, so the main additional parameters come from lightweight routers instead of multiple complete expert copies. To adapt the model to sparse routed inference, we perform Expert-Frozen Tuning as a cold-start phase of MoE Adaptation, where experts remain fixed while the router and other parameters are trainable. This stage preserves pretrained expert knowledge while establishing an initial routing policy. We then release the freezing constraint for full MoE Adaptation.

# 5 Experiments

## 5.1 Experimental Setup

**Models**  We evaluate three representative open-source unified models: BAGEL (Deng et al., 2025), Ming-Omni (AI et al., 2025), and Qwen-Image (Wu et al., 2025). For the understanding component, BAGEL and Qwen-Image rely on a VLM derived from Qwen-Instruct (Yang et al., 2024), whereas Ming-Omni employs an MoE-based backbone (Team, 2025). Their generation components differ more substantially: BAGEL employs a Mixture-of-Transformers (MoT) (Liang et al., 2025) design and reuses an LLM decoder for generation; Qwen-Image incorporates an MMDiT-based generator (Esser et al., 2024); and Ming-Omni adopts a multi-scale DiT block architecture. Table 1 compares these model families and lists the generator-side MLP blocks targeted by our component analysis.

We vary the number of experts per MoE layer among 16, 32, and 64. Following (Dai et al., 2024; DeepSeek-AI et al., 2025), we include shared experts that constitute one-sixteenth of the total number of experts. The

**Table 1:** Summary of evaluated unified models, component sizes, and generator-side MLP targets.

| Model | Understanding backbone | Understanding params | Generation backbone | Generation params | Generator MLP targets |
|---|---|---|---|---|---|
| Qwen-Image | Dense VLM | 7.62B | diffusion transformer | 20.42B | img_mlp, txt_mlp |
| Ming-Omni | MoE VLM | 17.12B | diffusion transformer | 2.51B | ff, ff_context, connector MLP |
| BAGEL | Dense VLM | 7.62B | autoregressive decoder | 7.62B | generation MLP stack |

overall activation ratio is set to 50% per layer. We exclude the first and last layers from MoE conversion, as they are important for preserving input encoding and output generation quality (Dai et al., 2024).

**Datasets** For calibration in training-free compression, we use a small number of examples drawn from the target task, which is sufficient for training-free compression (Frantar et al., 2022; Lin et al., 2023; Men et al., 2024; He et al., 2024). Note that these samples are used solely to compute activation scores, so no ground-truth annotations are required and no label leakage occurs. When calibration samples are drawn from the same benchmark prompt distribution as the evaluation set, we explicitly treat this as a task-calibrated or transductive compression protocol rather than as a standard held-out generalization setting. For MoE adaptation, we additionally incorporate high-quality image–text pairs, complemented by a small amount of synthetic data generated by existing text-to-image models.

Concretely, each task uses eight label-free calibration samples drawn from the same domain. For MoE Adaptation, each converted MLP layer has one router with per-token top-$k$ routed expert selection. The exact total/shared/top-$k$ settings are reported with the corresponding result tables.

Unless otherwise specified, the main BAGEL adaptation uses 16 total experts with one shared expert and top-7 routed experts per token, i.e., the 15/1/7 total/shared/top-$k$ setting, and converts layers 1–27 with an approximately 50% active-neuron budget. We use relaxed gates, $(1 + \text{Router}(x))$, initialized at zero so sparse layers start close to dense computation. Adaptation uses AdamW with a constant scheduler, betas 0.9/0.95, epsilon $10^{-15}$, weight decay 0, and gradient clipping 1.0. Router/expert-frozen tuning uses learning rate $10^{-8}$; full MoE Adaptation uses $10^{-5}$ and an auxiliary load-balancing loss.

## 5.2 Redundancy of Understanding Components

**Depth Reduction Works for Generation but Fails for Understanding** Because the understanding component is not directly responsible for the generated pixels, we first examine its effect on generation performance through depth reduction. Specifically, we remove transformer blocks, MLP layers, and attention layers. As shown in Figure 4, removing 50% of layers from the understanding component is effective for BAGEL and Qwen-Image, but less effective for Ming-Omni. We attribute this difference to architecture: Ming-Omni's generation component is relatively smaller and therefore depends more heavily on precise features encoded by the understanding component.

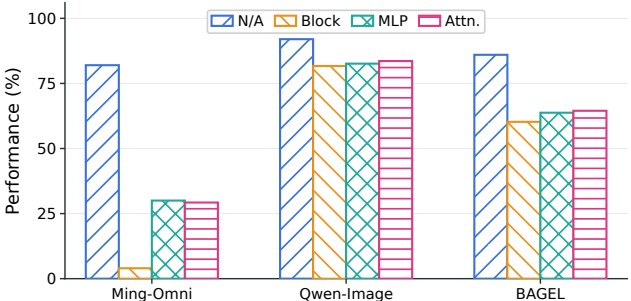

**Figure 4:** Overall GenEval performance after depth reduction in the understanding component.

However, depth pruning substantially degrades understanding capability. For instance, removing half of the MLP layers causes performance on MME (Fu et al., 2023) to drop from 1684.8 to 304.5 in perception and from 696.7 to 127.1 in cognition. These results suggest that depth reduction fails to preserve the performance of unified multimodal models on understanding tasks. Understanding tasks rely on autoregressive decoding, which is inherently an error-accumulation process: deviations in early generated tokens propagate through subsequent steps and ultimately cause the output to collapse.

**Table 2: GenEval performance after applying Neuron Partition to the understanding component**. Since only the **understanding component** is compressed, the reported parameter counts and speedups correspond to this part rather than the full model size.

| Model | Sparsity | Params. | Speedup | Single Obj. | Two Obj. | Counting | Colors | Position | Color Attri. | Overall↑ |
|---|---|---|---|---|---|---|---|---|---|---|
| BAGEL | 0% | 7.62B | 1.00× | 0.99 | 0.94 | 0.81 | 0.95 | 0.72 | 0.77 | **0.86** |
| | 50% | 4.76B | 1.60× | 0.94 | 0.63 | 0.62 | 0.77 | 0.47 | 0.34 | **0.63** |
| Qwen-Image | 0% | 7.62B | 1.00× | 0.99 | 0.98 | 0.91 | 0.94 | 0.80 | 0.89 | **0.92** |
| | 50% | 4.76B | 1.60× | 0.99 | 0.94 | 0.94 | 0.93 | 0.76 | 0.87 | **0.90** |
| | 70% | 3.62B | 2.10× | 0.97 | 0.88 | 0.85 | 0.91 | 0.60 | 0.71 | **0.82** |
| Ming-Omni | 0% | 17.12B | 1.00× | 0.97 | 0.95 | 0.67 | 0.92 | 0.71 | 0.71 | **0.82** |
| | 50% | 8.55B | 2.00× | 0.97 | 0.92 | 0.66 | 0.89 | 0.61 | 0.70 | **0.79** |
| | 70% | 5.61B | 3.05× | 0.96 | 0.81 | 0.58 | 0.86 | 0.49 | 0.56 | **0.71** |

**Table 3:** Performance of neuron partitioning on understanding tasks at compression ratios of 25% and 50%.

| Model | Sparsity | MME-P | MME-C | MMMU | MMBench | MMVP |
|---|---|---|---|---|---|---|
| Qwen-Image | – | 1711.6 | 611.8 | 50.0 | 87.8 | 75.7 |
| | 25% | 1574.6 | 458.2 | 39.7 | 83.9 | 70.3 |
| | 50% | 1165.9 | 271.1 | 29.9 | 77.4 | 62.0 |
| Ming-Omni | – | 1584.3 | 670.4 | 66.7 | 86.7 | 54.6 |
| | 25% | 1578.5 | 560.4 | 56.7 | 81.2 | 51.3 |
| | 50% | 1269.0 | 317.9 | 51.7 | 81.0 | 46.0 |
| BAGEL | – | 1684.8 | 696.7 | 65.0 | 88.1 | 69.6 |
| | 25% | 1558.1 | 681.7 | 60.1 | 85.7 | 68.7 |
| | 50% | 1392.6 | 528.9 | 56.7 | 79.2 | 56.0 |

**Neuron Partition on Understanding Components: Effective in Both Understanding and Generation** We next evaluate neuron partition on understanding components by compressing MLP layers to target ratios using a small calibration set. As shown in Table 2, Ming-Omni and Qwen-Image largely maintain performance even at aggressive compression ratios (50% and 70%), whereas BAGEL shows a larger degradation. This difference is likely due to BAGEL's mixture-of-transformers architecture (Liang et al., 2025), in which components interact through cross-attention at every layer. Similarly, the neuron partition strategy can be extended to attention heads, and this extension remains effective when compressing the understanding component for generation tasks (see Appendix B).

As shown in Table 3, neuron partition consistently outperforms layer dropping across understanding tasks. Although some layers exhibit substantial redundancy, removing entire layers also discards the smaller set of weights that remain critical to task performance (Yu et al., 2025). In contrast, neuron partition preserves important neurons within each layer, yielding more fine-grained and reliable compression. Nevertheless, since the understanding component directly governs textual outputs, its compression ratio should remain more conservative in understanding tasks than in generation tasks.

**Calibration Data Affects the Activated Parameters** Neuron partition uses calibration samples to estimate neuron importance and prune neurons that are less critical for the target behavior. Figure 2 shows that understanding and generation families activate different neurons, suggesting that calibration data can change which weights are retained and therefore affect downstream performance. To assess how different calibration datasets affect parameter retention and downstream performance, we conduct an ablation study using samples from understanding tasks (MME) and generation tasks (GenEval).

We find that aligning calibration data with target tasks consistently improves performance. For instance, using generation samples for calibration degrades MMBench from 79.2 to 74.8. The generation results in Figure 5 further highlight this trend. When calibrated with image generation samples, the outputs remain faithful to the prompts, producing broccoli, scissors, dolphins, and fruit bowls with correct structures. In contrast, calibration with understanding samples introduces distortions and mismatches. This demonstrates that family-aligned calibration data yields better performance, while mismatched data degrades generation quality. The effect is especially important for unified models, where inputs and outputs can vary across modality combinations.

| Baseline | Comp. w/Gen. | Comp. w/Und. | Baseline | Comp. w/Gen. | Comp. w/Und. |
| --- | --- | --- | --- | --- | --- |

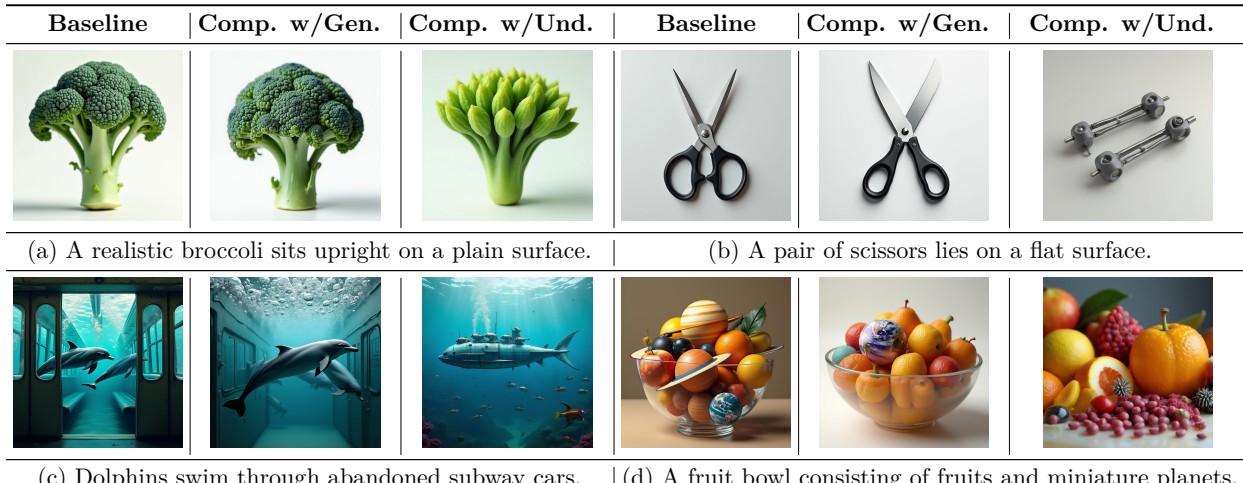

| (a) A realistic broccoli sits upright on a plain surface. | | | (b) A pair of scissors lies on a flat surface. | | |
| --- | --- | --- | --- | --- | --- |
| (c) Dolphins swim through abandoned subway cars. | | | (d) A fruit bowl consisting of fruits and miniature planets. | | |

**Figure 5: Impact of calibration data selection on neuron partition.** Each triplet shows the unmodified model, image-generation calibration, and understanding calibration.

| Baseline (uncompressed) | Compressed (50% width reduction) |
| --- | --- |

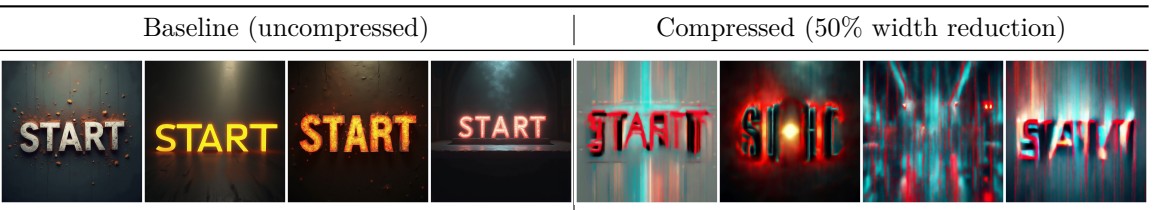

**Figure 6: Generation-side width reduction degrades visual quality.** The compressed model uses 50% generator width reduction for the prompt "The word START."

### 5.3 Dilemma of Generation Component Compression

We next investigate how compression influences generation quality by applying neuron partition to the generation components. Unlike understanding-side compression, generation-side compression presents a clear dilemma. Figure 6 shows that a 50% width reduction substantially degrades output fidelity and coherence, producing distorted structures and unrealistic textures.

These findings are consistent with the depth and attention reduction experiments in Appendix E. Together, they highlight a component-level asymmetry: understanding components can often tolerate compression, while generation components are highly sensitive to static compression.

Generation-side MLPs are also not homogeneous. Appendix Tables 12 and 13 separate the generator targets from Table 1 and report both their target FLOPs and their compression sensitivity. In Qwen-Image, compressing only `txt_mlp` retains much higher GenEval performance than compressing `img_mlp` (0.75 vs. 0.11), while compressing all generator MLPs falls to 0.05. Appendix Figure 13 provides a qualitative example of this text/context-side robustness. Ming-Omni remains broadly fragile, with the image-stream `ff` path collapsing to 0.00. Janus-Pro is treated as a scoped AR boundary case in Appendix Table 14: at matched keep 0.75, all AR-generation MLP compression collapses to 0.00, AR-language MLP compression reaches 0.65, and generation-head compression reaches 0.79.

### 5.4 MoE Adaptation for Generation Component Sparsity

Static compression conflicts with the dynamic activation patterns required across tasks and samples, leading to substantial degradation in generation components. To mitigate this issue, we apply MoE Adaptation, which enables dynamic activation and restores representational capacity under sparse inference.

**Table 4: Performance across stages of MoE Adaptation.** We compare training-free Expert Partition, Expert-Frozen Tuning, and full MoE Adaptation. Activated parameters are reported as "Und. Param. & Gen. Param."

| Method | Adapt. Comp. | Activated Params. | Single Obj. | Two Obj. | Counting | Colors | Position | Color Attri. | Overall↑ |
|---|---|---|---|---|---|---|---|---|---|
| Baseline | Dense | 7.62B + 7.62B | 0.99 | 0.94 | 0.81 | 0.95 | 0.72 | 0.77 | **0.86** |
| Expert Partition | | | 0.90 | 0.70 | 0.49 | 0.74 | 0.53 | 0.34 | **0.62** |
| Dense Finetuning | *Gen.* | 7.42B + 4.96B | 0.97 | 0.88 | 0.75 | 0.91 | 0.67 | 0.71 | **0.82** |
| Expert-Frozen Tuning | | | 0.99 | 0.94 | 0.62 | 0.93 | 0.69 | 0.54 | **0.78** |
| MoE Adaptation | | | **0.99** | **0.95** | **0.85** | **0.95** | **0.75** | **0.79** | **0.88** |
| Expert Partition | | | 0.69 | 0.18 | 0.23 | 0.45 | 0.10 | 0.05 | **0.28** |
| Dense Finetuning | *Und. & Gen.* | 4.96B + 4.96B | 0.97 | 0.89 | 0.76 | 0.91 | 0.70 | 0.64 | **0.81** |
| Expert-Frozen Tuning | | | 0.94 | 0.63 | 0.62 | 0.77 | 0.47 | 0.34 | **0.63** |
| MoE Adaptation | | | **0.99** | **0.96** | **0.78** | **0.95** | **0.70** | **0.72** | **0.85** |

| Prompt | Baseline | Zero-shot w/o SE | Zero-shot w/ SE | E.F. Tuning | MoE Adapt. |
|---|---|---|---|---|---|
| A famous flower that symbolizes wealth in China. | | | | | |
| Traditional activity during Easter in Western countries. | | | | | |
| Old analog picture of parked car on side street, quiet night. | | | | | |
| A lone astronaut paints swirling galaxies onto a massive canvas in a vast space station. | | | | | |

**Figure 8: Visual comparison across stages of MoE Adaptation.** Columns show the dense baseline, zero-shot expert partition with or without shared experts, Expert-Frozen Tuning, and full MoE Adaptation.

**Warmup via Expert-Frozen Tuning**   MoE Adaptation begins with Expert-Frozen Tuning, a cold-start phase that trains the model to use partitioned experts effectively. This strategy mitigates catastrophic forgetting and encourages expert selection while preserving pretrained knowledge (Houlsby et al., 2019; Qiao & Mahdavi, 2024; He et al., 2025a).

We compare 16, 32, and 64 experts, with training loss curves shown in Figure 7. After a small number of expert-frozen tuning steps, the loss declines substantially, indicat-

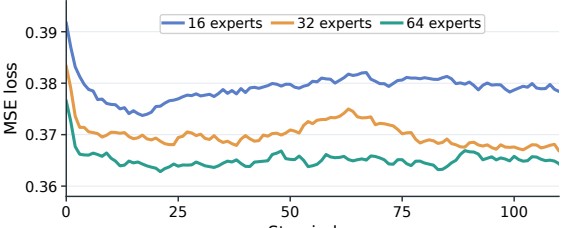

**Figure 7: Expert-Frozen Tuning curves.**

ing that the model can exploit a subset of experts to recover performance. Finer-grained expert partitioning also enables more flexible activation combinations, leading to lower training loss.

For instance, the overall GenEval score improves from 0.62 to 0.78, reflecting more coherent and visually faithful outputs. As illustrated in Figure 8, before tuning, the model generates noisy, low-detail images

**Table 5: Measured inference efficiency for dense, pruned, and MoE execution.** Rows use matched hardware, precision, workload, and converted-layer ranges within each model. MoE configs use total experts / shared experts / routed top-$k$. Memory is peak GPU memory; Appendix Table 9 reports target-component FLOPs separately.

| Model | Config | Dense | | | Pruned | | | MoE | | | MoE Speedup |
|---|---|---|---|---|---|---|---|---|---|---|---|
| | | Lat. | Thr. | Mem. | Lat. | Thr. | Mem. | Lat. | Thr. | Mem. | |
| Qwen-Image | 4e/1s/k1 | 54.438 s | 0.018/s | 63.36 GB | 45.089 s | 0.022/s | 62.93 GB | 44.970 s | 0.022/s | 63.14 GB | 1.211× |
| Ming-Omni | 8e/2s/k2 | 1.691 s | 0.591/s | 43.72 GB | 1.212 s | 0.825/s | 42.99 GB | 1.363 s | 0.734/s | 43.73 GB | 1.241× |
| BAGEL | 4e/1s/k1 | 28.196 s | 0.035/s | 25.101 GB | 25.589 s | 0.039/s | 20.723 GB | 25.892 s | 0.039/s | 25.068 GB | 1.089× |

that fail to capture fine-grained semantics. Expert-Frozen Tuning improves image fidelity and strengthens alignment between generated content and instructions. This indicates that subsets of generation parameters, although difficult to compress statically, can still support high-quality generation when routed and tuned appropriately.

**Effectiveness of MoE Adaptation**  After Expert-Frozen Tuning, we lift the restriction on frozen expert parameters to further improve performance. Beyond applying MoE adaptation only to the generation component, we also explore extending it to the understanding component to reduce the number of activated parameters while maintaining task effectiveness. To preserve fidelity in understanding tasks, the experts in the understanding component are kept frozen. They remain fully activated for understanding and are only sparsely activated for generation, since generation tasks are more tolerant to sparsity in this component. In summary, we consider two configurations of MoE adaptation: (1) *Gen.*: Expert partitioning and adaptation are applied only to generation experts; and (2) *Und. & Gen.*: Both understanding and generation experts are partitioned, but only the generation experts are adapted while the understanding experts remain frozen.

These additional training stages allow experts to refine their internal representations and develop stronger specialization, improving both structural coherence and semantic consistency in generated outputs. As shown in Table 4 and Figure 8, generation quality improves after Expert-Frozen Tuning and improves further after full MoE Adaptation. This demonstrates that allowing experts to adapt beyond static partitioning enhances both representational capacity and task alignment. We keep this quality comparison separate from inference speed measurements: adding a speedup column to Table 4 would require matched whole-generation timing for every checkpoint and adaptation scope, including both *Gen.* and *Und. & Gen.* partitioned settings.

**Efficiency Measurement**  MoE conversion targets MLP/FFN neurons, so target-component FLOPs provide a useful upper-bound context rather than a direct whole-model speed prediction. Appendix Table 9 reports these target-component FLOPs separately. The main runtime table focuses on measured end-to-end latency, throughput, and peak memory under matched dense, pruned, and MoE execution.

Table 5 reports measured latency, throughput, and peak memory for dense, pruned, and MoE execution. Latency is wall-clock time per generation workload, throughput is the reciprocal workload rate, and memory is end-to-end peak GPU memory. For Qwen-Image, the corresponding CUDA reserved memory is nearly unchanged between dense and MoE at the largest workload, 63.36 GB versus 63.14 GB, so the larger allocated-memory reduction reflects lower temporary/activation peaks rather than a smaller checkpoint footprint. The results show consistent end-to-end MoE speedups in the tested implementations. Pruned dense execution remains faster because it removes parameters without routing overhead. Together, the multi-model pruning results and runtime measurements support the broader component-level diagnosis, while Table 4 provides the training-aware MoE adaptation quality study on BAGEL GenEval. Appendix Table 9 gives the component-level target-FLOP context for these measurements, and Appendix Table 10 reports single-layer diagnostics that connect local sparse-compute savings to the end-to-end results.

We also evaluate the sensitivity of the expert design. Appendix F reports that BAGEL GenEval quality remains close across several total/shared/top-$k$ granularities at a fixed active-neuron budget. However, finer expert partitioning makes routed matrix multiplications smaller and more fragmented, which can be less favorable for kernel dispatch and practical optimization.

# 6 Conclusion

This work studies component-level redundancy and activation sparsity in unified multimodal models. Using training-free pruning as a probe, we find that the understanding component is substantially more compressible than the generation component, especially when it is used to support generation. In contrast, static compression of the generation component can severely degrade visual quality, indicating that generation relies on more input-dependent activation patterns. To address this asymmetry, we introduce Mixture-of-Experts (MoE) Adaptation, which dynamically activates generation parameters and restores generation quality under sparse activation on BAGEL. Together, the analysis and adaptation results show that unified multimodal models contain meaningful efficiency opportunities, but that effective compression must respect the distinct roles of understanding and generation components and be validated with measured runtime behavior.

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

**Appendix**

## A    Depth Reduction on Understanding Tasks

Depth reduction in the understanding component can have only a limited impact on some generation tasks, but it is brittle for multimodal understanding. Table 6 shows that removing whole layers sharply degrades MME, MMMU, MMBench, and MMVP. This is distinct from the neuron-partition results in Table 3, where fine-grained neuron selection is substantially more reliable.

Figure 9 illustrates the failure mode. The depth-reduced model starts to answer the visual question but quickly collapses into repetition, suggesting error accumulation during autoregressive decoding.

**Table 6:** Depth reduction degrades understanding performance. Whole-layer removal is less reliable than neuron partition.

| Model | Sparsity | MME-P | MME-C | MMMU | MMBench | MMVP |
|-------|----------|-------|-------|------|---------|------|
| Ming-Omni | – | 1584.3 | 670.4 | 66.7 | 86.7 | 54.6 |
|  | 50% | 1197.2 | 308.2 | 51.7 | 81.2 | 46.0 |
| Qwen-Image | – | 1711.6 | 611.8 | 50.0 | 87.8 | 75.7 |
|  | 50% | 7.0 | 1.4 | 10.6 | 12.8 | 30.3 |
| BAGEL | – | 1684.8 | 696.7 | 65.0 | 88.1 | 69.6 |
|  | 50% | 304.5 | 127.1 | 16.7 | 18.6 | 23.1 |

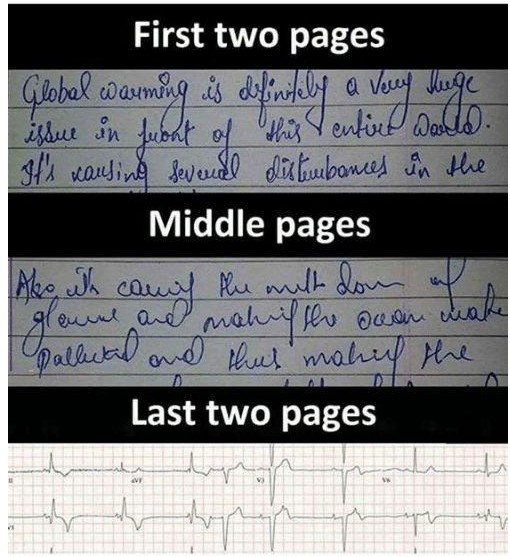

Meme: "My Handwriting In Exams"

**Question:** Can someone explain what's funny about this meme?

**Neuron Partition:**
The joke comes from the exaggerated shift in handwriting style as the exam progresses. The text starts out neat and legible, then becomes increasingly messy, reflecting the student becoming more stressed and less focused.

**Depth Reduction:**
The depth-reduced model produces degenerate repetition instead of a coherent explanation, making the answer unreadable and unhelpful.

**Figure 9:** Neuron partition preserves coherent answers, while depth reduction causes repetition. Both methods use a 50% compression ratio.

## B    Attention Head Partition

Our main experiments focus on MLP width reduction because MLP layers account for most parameters. The same score construction can also be applied to attention heads by measuring head-level importance. Table 7 shows that attention-head partition is still viable on the understanding component, although the quality drop is larger than for MLP neuron partition.

**Table 7:** Attention-head partition on the understanding component at 50% sparsity.

| Model | Compressed Layers | Single Obj. | Two Obj. | Counting | Colors | Position | Color Attri. | Overall↑ |
|-------|-------------------|-------------|----------|----------|--------|----------|--------------|----------|
| | none | 0.99 | 0.94 | 0.81 | 0.95 | 0.72 | 0.77 | 0.86 |
| BAGEL | 3-27 | 0.97 | 0.87 | 0.66 | 0.88 | 0.33 | 0.31 | 0.67 |
| | 4-27 | 0.98 | 0.91 | 0.72 | 0.89 | 0.41 | 0.40 | 0.72 |

## C  Comparison with Compression Methods

Neuron partition uses reconstruction errors to estimate neuron importance without labels or gradients. As a reference point, Table 8 compares it with gradient-based pruning metrics in the style of LLM-Pruner (Ma et al., 2023) and with AWQ 4-bit quantization. Neuron partition is competitive while remaining training-free and avoiding explicit gradient computation.

**Table 8:** Neuron partition compared with gradient-based pruning and 4-bit quantization on the understanding component.

**(a) Gradient-based pruning comparison.**

| Model | Metrics | Single Obj. | Two Obj. | Counting | Colors | Position | Color Attri. | Overall↑ |
|-------|---------|-------------|----------|----------|--------|----------|--------------|----------|
| Ming-Omni | LLM-Pruner | 0.96 | 0.82 | 0.72 | 0.85 | 0.47 | 0.55 | 0.70 |
| | Neuron Partition | 0.96 | 0.81 | 0.58 | 0.86 | 0.49 | 0.56 | 0.71 |

**(b) Quantization comparison.**

| Model | Single Obj. | Two Obj. | Counting | Colors | Position | Color Attri. | Overall↑ |
|-------|-------------|----------|----------|--------|----------|--------------|----------|
| Qwen-Image | 0.99 | 0.98 | 0.91 | 0.94 | 0.80 | 0.89 | 0.92 |
| w/ 4-bit quantization | 0.99 | 0.97 | 0.93 | 0.92 | 0.79 | 0.70 | 0.88 |
| w/ 50% neuron partition | 0.99 | 0.94 | 0.94 | 0.93 | 0.76 | 0.87 | 0.90 |

For Qwen-Image, we apply AWQ to the understanding component, namely the Qwen-VL backbone. Neuron partition reaches 0.90 GenEval overall at 50% compression, compared with 0.88 for 4-bit quantization. This contrasts with traditional LLM compression settings, where pruning at similar ratios is often substantially weaker than 4-bit quantization.

## D  Calibration Data for Neuron Partitioning

Calibration data determine which activations are measured and therefore which neurons are retained. We ablate this effect using MME samples for understanding calibration and GenEval samples for generation calibration. The resulting models are evaluated on MME and GenEval in Figures 10a and 10b.

Task-aligned calibration consistently performs better. This suggests that unified models rely on different neuron partitions for different modality families. It also explains why our test-time few-shot compression can adapt to downstream tasks by using a few target-task samples for calibration.

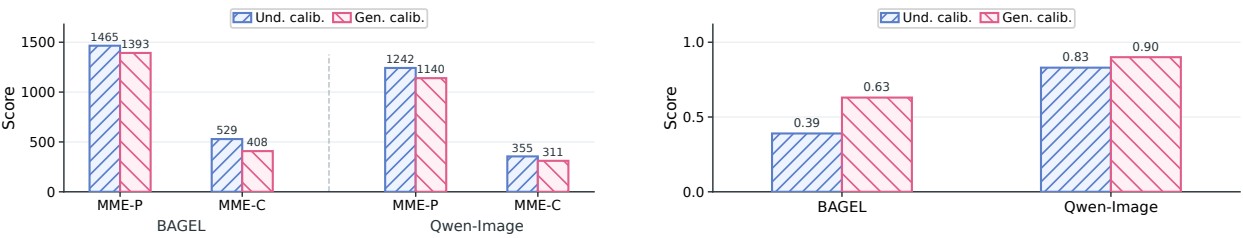

**(a)** Understanding tasks.         **(b)** Generation tasks.

**Figure 10:** Family-aligned calibration consistently yields better performance.

## E   Additional Generation Component Compression

Generation components are more sensitive to static compression than understanding components. Beyond neuron partition, we examine depth reduction and attention-layer compression. Removing entire generation layers produces severe visual degradation, as shown in Figure 11. Attention compression is also fragile: reducing more than 10% of attention capacity leads to visible quality drops in Figure 12.

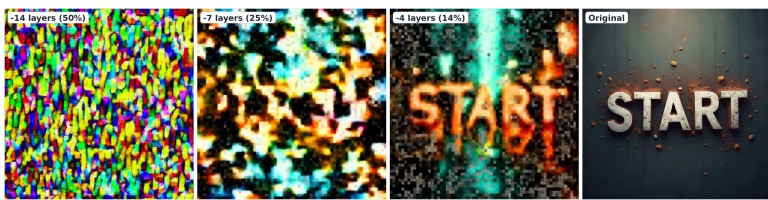

**Figure 11: Generation depth reduction.** Removing 14, 7, or 4 generation layers progressively degrades the generated images.

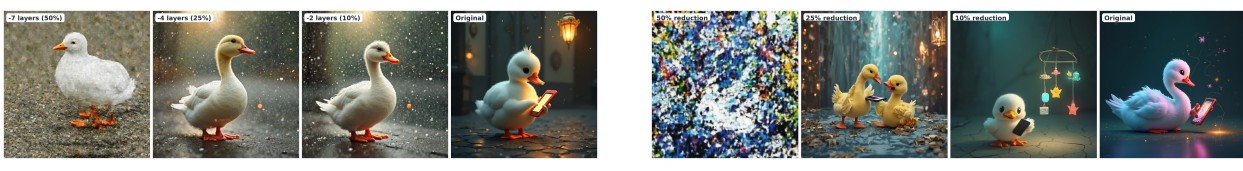

(a) Depth reduction.                                    (b) Width reduction.

**Figure 12: Attention-layer compression in generation components.** Both depth and width reduction substantially degrade generation quality.

## F   Additional System Efficiency Results

Table 9 reports the component-level compute context for the MLP/FFN blocks targeted by our compression and sparse-routing experiments. FLOPs count only MLP/FFN matrix multiplications. They should therefore be read as target-component compute rather than full-pipeline compute, which also includes attention, image decoders, sampling, routing, and framework overhead.

We also check expert-count sensitivity for BAGEL MoE Adaptation under a matched active-neuron budget. The default quality setting is 15/1/7, corresponding to 16 total experts with one shared expert and top-7 routed experts. GenEval remains close across the tested total/shared/top-$k$ settings, staying around 0.89 rather than improving with finer expert partitioning. This supports quality robustness, but it also exposes a systems trade-off: more experts make the routed matrix multiplications smaller and more fragmented, which is harder for GPU kernels and dispatch paths to optimize.

**Table 9: Component-level MLP/FFN compute context.** FLOPs are target-component FLOPs only, not whole-model FLOPs; activated FLOPs use the listed sparse method. MoE configs use total experts / shared experts / routed top-$k$.

| Model | Component family | Target params | Target dense FLOPs | Sparse method | Activated target FLOPs | E2E speedup |
|-------|------------------|---------------|--------------------|---------------|------------------------|-------------|
| BAGEL | image-decoder MLP stack | 5.70B | 186.88 TF | MoE 4e/1s/k1 | 93.44 TF | 1.089× |
| Qwen-Image | MMDiT image/text FFNs | 9.06B | 765.36 TF | MoE 4e/1s/k1 | 306.14 TF | 1.211× |
| Ming-Omni | multi-scale DiT FFNs | 0.90B | 36.84 TF | MoE 8e/2s/k2 | 14.74 TF | 1.241× |

Table 10 complements the end-to-end timing in Table 5 with single-layer latency diagnostics under matched token counts and sparse FLOPs. These rows are not whole-model generation timings: BAGEL is measured on one generation MLP layer, while Qwen-Image and Ming-Omni are measured on one MMDiT FFN layer. BAGEL has the largest per-layer MLP workload in this diagnostic, so routing overhead is amortized better and its single-layer MoE speedup is closer to the pruned upper bound.

Table 11 reports the corresponding single-layer memory diagnostics. Qwen-Image and Ming-Omni report resident FFN-stack memory after loading the stack and input. The BAGEL diagnostic records CUDA allocator peak memory for the token-matched single-layer run, so it is included as a local allocator-peak check rather than as a resident-footprint row.

**Table 10: Single-layer FFN/MLP latency diagnostics.** These module-level rows are not whole-model generation timings; sparse variants are matched by target FLOPs.

| Model / scope | Module | Tokens | Dense baseline | Method | Configuration | Sparse FLOPs | Latency | Speedup |
|---|---|---|---|---|---|---|---|---|
| BAGEL single layer | generation MLP | 32,768 | 74.185 ms | Pruned | 50% width | 4.43 TF | 36.336 ms | 2.042× |
| | | | | MoE | 3e/1s/k1 | 4.43 TF | 38.983 ms | 1.903× |
| | | 65,536 | 150.980 ms | Pruned | 50% width | 8.86 TF | 73.524 ms | 2.053× |
| | | | | MoE | 3e/1s/k1 | 8.86 TF | 77.830 ms | 1.940× |
| Qwen-Image single layer | MMDiT FFN | 32,768 | 27.395 ms | Pruned | 50% width | 2.47 TF | 13.791 ms | 1.986× |
| | | | | MoE | 3e/1s/k1 | 2.47 TF | 17.127 ms | 1.600× |
| | | 65,536 | 54.206 ms | Pruned | 50% width | 4.95 TF | 28.852 ms | 1.879× |
| | | | | MoE | 3e/1s/k1 | 4.95 TF | 35.648 ms | 1.521× |
| Ming-Omni single layer | MMDiT FFN | 32,768 | 7.405 ms | Pruned | 50% width | 0.62 TF | 3.428 ms | 2.160× |
| | | | | MoE | 3e/1s/k1 | 0.62 TF | 5.204 ms | 1.423× |
| | | 65,536 | 14.875 ms | Pruned | 50% width | 1.24 TF | 7.706 ms | 1.930× |
| | | | | MoE | 3e/1s/k1 | 1.24 TF | 11.580 ms | 1.285× |

**Table 11: Single-layer memory diagnostics.** Qwen-Image and Ming-Omni report resident FFN-stack memory; BAGEL reports CUDA allocator peak memory for the same token-matched layer diagnostic.

| Model / scope | Memory type | Tokens | Dense | Pruned | MoE |
|---|---|---|---|---|---|
| BAGEL | peak | 32,768 | 34.042 GB | 31.479 GB | 30.346 GB |
| BAGEL | peak | 65,536 | 39.226 GB | 34.295 GB | 31.832 GB |
| Qwen-Image | resident | 32,768 | 8.630 GB | 4.410 GB | 8.630 GB |
| Qwen-Image | resident | 65,536 | 8.810 GB | 4.590 GB | 8.820 GB |
| Ming-Omni | resident | 32,768 | 0.940 GB | 0.520 GB | 0.940 GB |
| Ming-Omni | resident | 65,536 | 1.030 GB | 0.610 GB | 1.030 GB |

## G  Additional Generation-Component Compression Results

**Generator-subcomponent compression.** The generator-side target is not homogeneous across models. BAGEL uses a diffusion-style image generator whose main converted target is the image-decoder MLP stack. Qwen-Image uses a dual-stream MMDiT with separate image-stream and text/context-stream MLPs. Ming-Omni separates image, context, and connector MLPs. Table 12 breaks down the dominant generator-side MLP/FFN FLOPs for Qwen-Image and Ming-Omni. Janus-Pro uses a shared AR language backbone together with a generation head and aligner, so we report it as an applicability boundary rather than part of the MoE efficiency table.

**Table 12: Generator-subcomponent compute context.** These rows identify which MLP/FFN subcomponents dominate target compute.

| Model | Component | Role | Params | Dense FLOPs |
|---|---|---|---|---|
| Qwen-Image | img_mlp | image stream | 4.53B | 626.21 TF |
| | txt_mlp | text/context stream | 4.53B | 139.16 TF |
| Ming-Omni | ff | image stream | 0.45B | 27.83 TF |
| | ff_context | context stream | 0.43B | 8.75 TF |
| | connector MLP | LLM-to-diffusion connector | 0.01B | 0.25 TF |

We further isolate generation-side MLP streams with 50% structural compression. Table 13 shows that text/context-side generator MLPs can be less sensitive than image-stream MLPs, although the degree of robustness is model-dependent. Qwen-Image preserves much more GenEval performance when only `txt_mlp` is compressed, and Figure 13 gives a qualitative example under stronger text/context compression. Ming-Omni remains broadly fragile, but `ff_context` is still less damaging than the latent/image `ff` stream. Janus-Pro is reported separately in Table 14, where the generation-head compression ratio is aligned with the AR-language MLP setting.

Table 13: **Generator-subcomponent sensitivity at 50% compression.** GenEval is measured after compressing only the listed component.

| Model | Compressed component | Role | GenEval |
|---|---|---|---|
| Qwen-Image | all generator MLPs | image + text streams | 0.05 |
| | img_mlp | image stream | 0.11 |
| | txt_mlp | text/context stream | **0.75** |
| Ming-Omni | all generator MLPs | image + context streams | 0.00 |
| | ff | image stream | 0.00 |
| | ff_context | context stream | **0.19** |
| | ff+ff_context | image + context streams | 0.00 |
| | VAE decoder conv | decoder convolution | 0.02 |

| Dense
0% comp. | txt
50% comp. | txt
70% comp. | txt
75% comp. | text_encoder+txt
75% comp. |
|---|---|---|---|---|

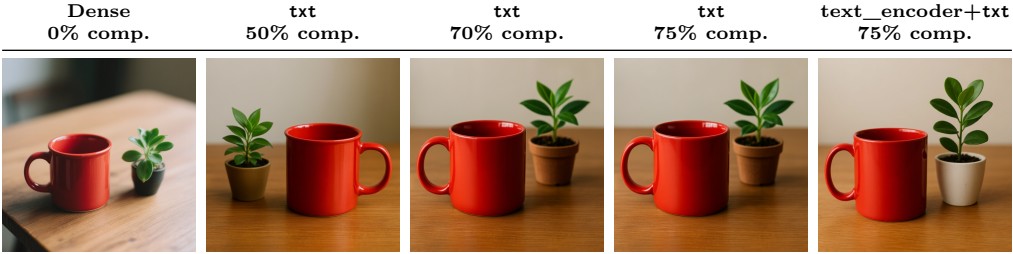

Figure 13: **Qwen-Image `txt_mlp` compression.** Percentages denote the compressed fraction.

**Autoregressive generator compression.** Table 14 gives a scoped Janus-Pro AR-generation ablation. It is not a direct MoE Adaptation claim because Janus-Pro shares an AR language backbone for understanding and image-token generation. We compare overall AR-generator compression, AR-language MLP compression, and generation-head compression at matched 25% compression. Compressing the full AR-generation MLP target remains destructive, while the AR-language MLP and generation head behave differently under the same keep ratio.

Table 14: **Janus-Pro AR GenEval.** Rows report matched keep and compression ratios.

| Target | Keep | Comp. | Score | Ret. |
|---|---|---|---|---|
| dense baseline | 100% | 0% | 0.76 | 100% |
| all AR-gen MLPs | 75% | 25% | 0.00 | 0% |
| AR language MLP | 75% | 25% | 0.65 | 86% |
| generation head | 75% | 25% | **0.79** | **104%** |

## H   Limitations and Broader Impact

**Limitations**   Our pruning and partition diagnostics cover BAGEL, Qwen-Image, and Ming-Omni; trained MoE Adaptation is a BAGEL/GenEval case study. Extending it to other generators remains future work. Activated parameters are not deployment speedup by themselves: latency also depends on workload size, expert granularity, routing overhead, kernels, and hardware.

**Broader Impact**   Improving unified-generator efficiency can reduce serving cost and broaden access, but may also lower the barrier for large-scale synthetic-content generation. Risks include misinformation, deepfake-like misuse, biased or harmful outputs, and copyright-sensitive generation. Deployment should pair efficiency gains with provenance or watermarking, safety filters, dataset filtering, and careful release practices.

