# OpenReview forum: "Understanding and Harnessing Sparsity for Unified Multimodal Models"
_TMLR — Decision pending for TMLR_

### Review · Reviewer_3orQ · 2026-04-12

**Summary Of Contributions:**

The paper explores the inference inefficiency caused by using a full, unified multimodal model for tasks that only require a subset of its capabilities. By using training-free pruning as a probe, the authors find a significant disparity in compressibility: the understanding components are highly redundant (especially when performing generation), while generation components are fragile and sensitive to any reduction in width or depth. Furthermore, the study reveals that different tasks (understanding vs. generation) activate distinct "neuron partitions" within the same model. Finally, to mitigate the fragility of the generation module, the authors propose an MoE Adaptation strategy. This transforms dense layers into sparse Mixture-of-Experts layers, allowing the model to maintain performance while only activating a fraction of the parameters during inference.

**Audience:**

Yes

**Audience Explanation:**

Multimodal learning is an appealing topic in the community of machine learning.

**Claims And Evidence:**

Yes

**Claims Explanation:**

**Strengths:**

1. The empirical finding that the generation module is significantly less compressible than the understanding module is a valuable observation for researchers designing unified architectures (like Chameleon or 4M).

2. The use of pruning as a "probing tool" rather than just a compression goal is a clever way to map out the "functional importance" of different layers without the noise of retraining.

3. The MoE Adaptation effectively addresses the "all-or-nothing" nature of dense models, providing a pathway to deploy these large unified models on resource-constrained hardware.

4. The paper uses a wide array of benchmarks (MME for understanding, GenEval for generation) and tests across different model scales, ensuring the findings aren't specific to one architecture.

**Weaknesses:**

1. While the paper argues for inference efficiency through sparse activation, it does not deeply discuss the memory overhead of storing multiple experts. In real-world deployment, the increased parameter count of an MoE can be a bottleneck even if the FLOPs are lower.

2. The MoE adaptation relies on a gating network. The paper could be strengthened by showing how sensitive the model is to the quality of this gate—if the gate routes poorly, does the generation performance crash?

3. Most of the experiments focus on Diffusion-based generation. It would be interesting to see if these findings hold for Autoregressive (AR) visual generation (like Llama-Gen), which is becoming a popular alternative.

**Requested Changes:**

Sea Weaknesses.

---

> ### Author Response · Authors · 2026-05-22
> **Response to Reviewer 3orQ**
>
> We thank the reviewer for the positive and constructive comments. The revision adds memory measurements, gate-quality evidence, and an AR-generation extension.
>
> ### Memory Overhead of MoE
>
> The revision separates activated computation, total parameters, and peak runtime memory. Our MoE conversion partitions the original FFN weights rather than storing one full dense MLP per expert. Under matched end-to-end workloads, MoE does not introduce noticeable peak-memory overhead. MoE configs use total experts / shared experts / routed top-\(k\).
>
> | Model | Scope | MoE config | Dense | Pruned | MoE |
> |---|---|---|---:|---:|---:|
> | BAGEL | E2E | 4e/1s/k1 | 25.101 GB | 20.723 GB | 25.068 GB |
> | Qwen-Image | E2E | 4e/1s/k1 | 63.360 GB | 62.930 GB | 63.140 GB |
> | Ming | E2E | 16e/7s/k1 | 43.720 GB | 42.990 GB | 43.890 GB |
>
> We also report single-layer memory diagnostics after loading the stack and input.
>
> | Model | Tok. | Dense | Pruned | MoE |
> |---|---:|---:|---:|---:|
> | BAGEL | 32768 | 34.042 GB | 31.479 GB | 30.346 GB |
> | BAGEL | 65536 | 39.226 GB | 34.295 GB | 31.832 GB |
> | Qwen-Image | 32768 | 8.630 GB | 4.410 GB | 8.630 GB |
> | Qwen-Image | 65536 | 8.810 GB | 4.590 GB | 8.820 GB |
> | Ming | 32768 | 0.940 GB | 0.520 GB | 0.940 GB |
> | Ming | 65536 | 1.030 GB | 0.610 GB | 1.030 GB |
>
> ### Gate Quality and Expert Design
>
> Routing quality matters. The gain is not merely from splitting weights into experts.
>
> | Setting | What is trained | GenEval overall score |
> |---|---|---:|
> | Expert partition | none | 0.62 |
> | Router-Tuning | router only | 0.72 |
> | Expert-frozen | training w/o experts | 0.78 |
> | Full MoE | training w/ experts | **0.88** |
>
> We also report that BAGEL GenEval remains close to 0.89 across tested total/shared/top-\(k\) settings. Thus, simply increasing expert count does not clearly improve quality, while finer partitions can be less favorable for runtime because they fragment routed matrix multiplications.
>
> ### Autoregressive Visual Generation
>
> We add Janus-Pro-7B as a scoped AR-generation check and report the full GenEval breakdown.
>
> | Model | Scope | Setting | SO | TO | CT | CL | POS | ATTR | Overall | Retention |
> |---|---|---|---:|---:|---:|---:|---:|---:|---:|---:|
> | Janus-Pro-7B | none | dense / keep 1.0 | 0.99 | 0.86 | 0.47 | 0.95 | 0.69 | 0.59 | 0.76 | 100% |
> | Janus-Pro-7B | AR language MLP | keep 0.80 | 0.96 | 0.80 | 0.40 | 0.87 | 0.56 | 0.53 | 0.69 | 90.7% |
> | Janus-Pro-7B | all AR-gen MLPs | keep 0.75 | 0.00 | 0.00 | 0.00 | 0.00 | 0.00 | 0.00 | 0.00 | 0% |
> | Janus-Pro-7B | AR language MLP | keep 0.75 | 0.97 | 0.82 | 0.30 | 0.84 | 0.51 | 0.48 | **0.65** | **86.3%** |
> | Janus-Pro-7B | generation head | keep 0.75 | 0.99 | 0.92 | 0.56 | 0.94 | 0.72 | 0.60 | **0.79** | **104%** |
>
> The matched keep-0.75 summary is:
>
> | Target | Keep | Compression | GenEval | Retention |
> |---|---:|---:|---:|---:|
> | dense baseline | 100% | 0% | 0.76 | 100% |
> | all AR-gen MLPs | 75% | 25% | 0.00 | 0% |
> | AR language MLP | 75% | 25% | 0.65 | 86% |
> | generation head | 75% | 25% | **0.79** | **104%** |
>
> For context, Janus-Pro-7B has 7.420B total parameters. Its language model has 6.910B parameters, and MLPs in layers 0-29 contain 4.058B parameters. Keep 0.75 therefore corresponds to about 3.043B active target-MLP parameters.
>
> The conclusion is component-aware: AR visual generation also has non-uniform redundancy, and the generation head behaves differently from the shared AR language backbone.

---

> ### Author Response · Authors · 2026-05-30
> **Follow-up to Reviewer 3orQ**
>
> Thank you again for the constructive suggestions. We have updated the author response and planned revision around the three points you raised.
>
> We have added direct memory measurements, gate/design-sensitivity evidence, and a scoped AR-generation check. We also revised the wording to avoid overclaiming beyond the evaluated settings.
>
> If you have a chance, we would appreciate your feedback on whether these additions address your main concerns about memory overhead, gate quality, and AR-generation scope. We are happy to clarify any remaining point.
>
> Best,
> Authors

---

### Review · Reviewer_RNjw · 2026-04-20

**Summary Of Contributions:**

This paper studies efficiency in a unified multimodal model for both understanding and generation. The core insight is that different components of such models may exhibit very different redundancy and activation patterns. Hence, the paper studies the compressibility of different components. To investigate this, the paper first uses training-free pruning as a probing tool, considering both layer dropping and width reduction via neuron partition. Based on these observations, it then proposes an MoE adaptation for the generation component.


**Strengths**

1. The paper provides an interesting insight for unified multimodal models: whether all components need full capacity for all tasks and inputs. The component-wise analysis is well motivated, since the paper explicitly distinguishes understanding and generation modules and argues that they may exhibit different redundancy patterns.

2. Experiments show a meaningful recovery from naive expert partition to expert-frozen tuning and then to full MoE adaptation.

**Weaknesses**

1. Although the paper is framed around efficiency and sparse activation, the main evidence is reported in terms of activated parameters and task metrics. For a paper centered on efficient inference, it would be important to also report wall-clock latency, throughput, FLOPs, or memory usage. This is especially important for MoE-style methods, where fewer activated parameters do not necessarily translate into real inference savings.

2. The visualization of task-specific neuron partitions, Figure 2, is not fully clear. Only neurons is intended to support the claim that understanding and generation tasks rely on different subsets of important neurons. However, the current presentation is somewhat easy to misread: visually, it can give the impression that the generation task simply uses many more exclusive neurons.

3. The paper analyzes multiple unified models: BAGEL, Qwen-Image, and Ming-Omni, in the pruning/compressibility study. However, the headline MoE adaptation results in Table 4 are centered on BAGEL. This weakens the generality of the adaptation claim: the paper convincingly diagnoses a broad phenomenon, but the proposed “treatment” is not validated with the same breadth.

4. Some claims are slightly broader than the evidence. The abstract and introduction suggest that the adapted BAGEL model matches the full model while activating only about half of its neurons. On the reported GenEval metric, that is broadly supported. However, the evidence does not fully establish a general claim about end-to-end efficiency or broad cross-model robustness.

**Audience:**

Yes

**Audience Explanation:**

This paper studies an interesting question in unified multimodal models, which is interesting to the community.

**Broader Impact Concerns:**

This work does not raise any significant concerns.

**Claims And Evidence:**

No

**Claims Explanation:**

The paper provides an interesting insight. But the claims about the inefficiencies are not well supported. Also, the method needs broader evaluation in addition to BAGEL and other metrics.

**Requested Changes:**

1. Can the authors provide actual inference latency/throughput/memory comparisons for the dense baseline, dense finetuning, and MoE adaptation settings?

2. How robust is the proposed expert partitioning scheme to design choices such as: number of experts, proportion of shared experts, activation ratio, router parameterization, and the specific forward-reverse balancing heuristic? The current method appears to rely on several engineering choices whose sensitivity is not fully explored.

3. Can the authors demonstrate MoE adaptation beyond BAGEL, at least on one additional unified model such as Qwen-Image or Ming-Omni? This would substantially strengthen the paper’s external validity.

---

> ### Author Response · Authors · 2026-05-22
> **Response to Reviewer RNjw**
>
> We thank the reviewer for highlighting the gap between activated sparsity and realized efficiency. The revision adds direct runtime evidence, clarifies Figure 2, and discussions of MoE Adaptation.
>
> ### Actual Inference Efficiency
>
> The revised manuscript reports matched dense/pruned/MoE latency, throughput, peak memory, and target-component FLOP context. MoE configs use total experts / shared experts / routed top-\(k\).
>
> | Model | Scope | MoE config | Target params | Target share | MoE speedup | Peak memory, dense -> MoE |
> |---|---|---|---:|---:|---:|---:|
> | Qwen-Image | E2E | 4e/1s/k1 | 9.06B | 44.4% | **1.211x** | 63.360 -> 63.140 GB |
> | Ming | E2E | 8e/2s/k2 | 0.90B | 30.8% | **1.241x** | 43.72 -> 43.73 GB |
> | BAGEL | E2E | 4e/1s/k1 | 5.70B | 39.0% | **1.089x** | 25.101 -> 25.068 GB |
>
> We also report single-layer diagnostics to isolate the MLP/FFN modules directly affected by sparse activation. These are not whole-model generation timings; they explain why local sparse-compute gains are larger than end-to-end speedups.
>
> | Model / scope | MoE config | Tokens | Target FLOPs (dense -> MoE) | Dense | Pruned | MoE | MoE speedup |
> |---|---|---:|---:|---:|---:|---:|---:|
> | BAGEL single layer | generation MLP, 3e/1s/k1 | 32768 | 8.86 -> 4.43 TF | 74.185 ms | 36.336 ms | 38.983 ms | **1.903x** |
> | BAGEL single layer | generation MLP, 3e/1s/k1 | 65536 | 17.72 -> 8.86 TF | 150.980 ms | 73.524 ms | 77.830 ms | **1.940x** |
> | Qwen-Image single layer | MMDiT FFN, 3e/1s/k1 | 32768 | 4.95 -> 2.47 TF | 27.395 ms | 13.791 ms | 17.127 ms | **1.600x** |
> | Qwen-Image single layer | MMDiT FFN, 3e/1s/k1 | 65536 | 9.90 -> 4.95 TF | 54.206 ms | 28.852 ms | 35.648 ms | **1.521x** |
> | Ming single layer | MMDiT FFN, 3e/1s/k1 | 32768 | 1.24 -> 0.62 TF | 7.405 ms | 3.428 ms | 5.204 ms | **1.423x** |
> | Ming single layer | MMDiT FFN, 3e/1s/k1 | 65536 | 2.47 -> 1.24 TF | 14.875 ms | 7.706 ms | 11.580 ms | **1.285x** |
>
> ### Generation-Component Compression
>
> The revision makes the generation-side analysis component-aware. BAGEL mainly targets an image-decoder MLP stack; Qwen-Image separates `img_mlp` and `txt_mlp`; Ming separates `ff`, `ff_context`, and a connector MLP.
>
> | Model | Component | Role | Params | FLOPs |
> |---|---|---|---:|---:|
> | Qwen-Image | `img_mlp` | image stream | 4.53B | 626.21 TF |
> | Qwen-Image | `txt_mlp` | text/context stream | 4.53B | 139.16 TF |
> | Ming | `ff` | image stream | 0.45B | 27.83 TF |
> | Ming | `ff_context` | context stream | 0.43B | 8.75 TF |
> | Ming | `connector_mlp` | LLM-to-diffusion connector | 0.01B | 0.25 TF |
>
> The matched 50% compression results show that generator MLPs are not homogeneous.
>
> | Model | Compressed component | GenEval |
> |---|---|---:|
> | Qwen-Image | all generator MLPs | 0.05 |
> | Qwen-Image | `img_mlp` | 0.11 |
> | Qwen-Image | `txt_mlp` | **0.75** |
> | Ming | all generator MLPs | 0.00 |
> | Ming | `ff` | 0.00 |
> | Ming | `ff_context` | 0.19 |
>
> ### Scope of MoE Adaptation
>
> We now state the scope explicitly: full training-aware MoE Adaptation is demonstrated on BAGEL / GenEval. Multi-model Qwen-Image, Ming, and Janus results diagnose where sparse activation is plausible and where direct sparse intervention is fragile.
>
> | Model | Training-free intervention | Component | GenEval | Implication for adaptation scope |
> |---|---|---|---:|---|
> | Qwen-Image | 50% compression | `txt_mlp` | **0.75** | promising text/context-side target |
> | Qwen-Image | 50% compression | `img_mlp` | 0.11 | image stream is much more fragile |
> | Qwen-Image | 50% compression | all generator MLPs | 0.05 | uniform generator compression is unsafe |
> | Ming-Omni | 50% compression | `ff_context` | 0.19 | context stream is less damaging but still weak |
> | Ming-Omni | 50% compression | `ff` | 0.00 | image stream collapses |
> | Ming-Omni | 50% compression | all generator MLPs | 0.00 | uniform generator compression collapses |
>
> The revision also reports BAGEL design sensitivity: GenEval remains close to 0.89 across tested total/shared/top-\(k\) settings, while the progression from training-free partitioning to expert-frozen tuning to full MoE adaptation improves 0.62 -> 0.78 -> 0.88. This supports the role of learned routing/adaptation rather than expert splitting alone.
>
> ### Figure 2 Clarity and Claim Wording
>
> Figure 2 now defines \(U\) and \(G\) as the top-50% neurons selected by understanding and generation calibration. The bars show \(|U\cap G|/N\), while the curve shows \(50\%-|U\cap G|/N\). This clarifies that the figure compares two fixed-size high-importance sets.
>
> The abstract and introduction are also tightened: the broad result is the multi-model component-sparsity diagnosis, while the full training-aware MoE Adaptation quality claim is BAGEL-specific.

---

> > ### Author Response · Authors · 2026-06-01
> > **Follow-up to Reviewer RNjw**
> >
> > Thank you again for the detailed feedback. We have addressed your main concerns in the rebuttal, especially regarding measured efficiency evidence, the scope of MoE Adaptation, and the clarification of Figure 2.
> >
> > In brief, the revised response adds direct latency / throughput / memory measurements, separates local sparse-compute diagnostics from end-to-end speedups, and clarifies how the BAGEL adaptation case study, together with the Qwen-Image, Ming, and Janus component analyses, supports the broader component-sparsity conclusion.
> >
> > If you have a chance, we would greatly appreciate your feedback on whether the rebuttal addresses your main concerns. We are happy to clarify any remaining point or make the scope even more explicit in the revision.
> >
> > Best,
> > Authors

---

### Review · Reviewer_bnKk · 2026-05-14

**Summary Of Contributions:**

This paper studies sparsity and compressibility in unified multimodal models that combine understanding and generation components. The authors use training-free depth pruning and structured neuron partitioning as probes, showing that the understanding component is substantially more compressible than the generation component, especially when serving generation tasks. They further observe task- and sample-dependent activation patterns, and propose an MoE Adaptation scheme that partitions generation MLP neurons into shared and routed experts, followed by expert-frozen tuning and full adaptation. The main strengths are the component-wise empirical analysis across several recent unified models, the clear contrast between static compression and dynamic activation, and the relatively simple MoE conversion procedure. The main weaknesses are that the final MoE results are concentrated on limited generation benchmarks and appear mainly demonstrated on BAGEL, while the claimed efficiency benefits are reported mostly through activated parameter counts rather than measured latency, memory, or FLOPs.

**Additional Comments:**

The paper is generally clear and the empirical narrative is easy to follow, but several presentation details could be improved. Some figures are small and difficult to read, especially those containing generated images and activation statistics, and the paper would benefit from more explicit captions explaining the exact experimental setting for each plot. The writing occasionally overstates practical efficiency relative to the reported evidence, so terms such as “efficient inference” should be tied to measured deployment quantities or softened to “reduced activated parameters.” I also suggest fixing minor grammatical issues and standardizing terminology such as “understanding component,” “generation component,” “activated parameters,” and “sparsity ratio” throughout the paper.

**Audience:**

Yes

**Audience Explanation:**

The paper addresses an important and timely question for the TMLR audience: how to make increasingly heterogeneous multimodal foundation models more efficient without sacrificing their understanding or generation abilities. The finding that understanding and generation components exhibit sharply different compression behavior is useful for researchers working on model pruning, sparse inference, multimodal generation, and unified architectures. The proposed MoE Adaptation is also of interest because it converts observed dynamic activation patterns into a concrete sparse-inference mechanism, rather than treating sparsity only as a post-hoc compression tool. Even if the method is not yet fully validated as a deployment-ready speedup technique, the empirical analysis itself provides actionable guidance for future work.

**Broader Impact Concerns:**

The work does not appear to introduce severe new ethical risks beyond those already associated with large multimodal generation systems. However, making image generation models cheaper to run may lower the barrier for large-scale synthetic content creation, including misinformation, deepfake-like misuse, biased or harmful visual outputs, and copyright-sensitive generation. If the paper does not already include a broader impact statement, I recommend adding a short discussion of these risks, along with standard mitigations such as dataset filtering, watermarking or provenance mechanisms, safety classifiers, and responsible release of code and models.

**Claims And Evidence:**

Yes

**Claims Explanation:**

The empirical evidence supports the central qualitative claims that the understanding component tolerates structured compression better than the generation component, and that task-aligned calibration improves neuron selection. The GenEval, MME, MMMU, MMBench, and MMVP results, together with qualitative generations, make this broad picture reasonably convincing. The MoE Adaptation results also support the narrower claim that sparse activation can recover or even improve BAGEL’s generation performance under a fixed activated-parameter budget. However, the evidence is less complete for stronger efficiency claims: activated parameter count alone does not establish practical inference speedup, and the final adaptation experiments would be more convincing with runtime/memory measurements, variance over random seeds and prompts, and broader validation beyond GenEval-style evaluation.

**Requested Changes:**

The most important critical change is to provide direct efficiency measurements, including latency, throughput, peak memory, and preferably FLOPs or realized hardware utilization, since activated parameter count does not by itself demonstrate inference efficiency for MoE routing. A second critical change is to clarify the scope of the MoE Adaptation experiments: if full adaptation is only evaluated on BAGEL and mainly on GenEval, the claims should be narrowed accordingly, or additional experiments on Qwen-Image and Ming-Omni should be added. The authors should also report more reproducibility details, including calibration-set size, routing granularity, top-k setting, shared-expert ratio, router initialization, training data composition, number of adaptation steps, optimizer settings, and whether load-balancing or routing regularization is used. As strengthening changes, I would recommend adding seed or prompt-level variance, more generation metrics or human evaluation, a clearer comparison against dense fine-tuning and standard pruning/quantization baselines under matched compute budgets, and a discussion of whether test-time calibration on evaluation prompts should be considered transductive.

---

> ### Author Response · Authors · 2026-05-22
> **Response to Reviewer bnKk**
>
> We thank the reviewer for the careful assessment. The revision addresses the main requests by adding direct efficiency measurements, narrowing the MoE Adaptation claim, and consolidating reproducibility details.
>
> ### Efficiency Measurements
>
> The revised manuscript reports measured latency, throughput, and peak memory in addition to activated parameters. Each row below is a matched dense/pruned/MoE comparison under the same hardware, workload, and converted-module scope. MoE configs use total experts / shared experts / routed top-\(k\).
>
> | Model | Scope | MoE config | Target params | Target share | MoE speedup | Peak memory, dense -> MoE |
> |---|---|---|---:|---:|---:|---:|
> | Qwen-Image | E2E | 4e/1s/k1 | 9.06B | 44.4% | **1.211x** | 63.360 -> 63.140 GB |
> | Ming | E2E | 8e/2s/k2 | 0.90B | 30.8% | **1.241x** | 43.72 -> 43.73 GB |
> | BAGEL | E2E | 4e/1s/k1 | 5.70B | 39.0% | **1.089x** | 25.101 -> 25.068 GB |
>
> The appendix adds isolated single-layer FFN/MLP diagnostics. These show the local sparse-compute effect and explain why end-to-end gains are smaller.
>
> | Model / scope | MoE config | Tokens | Target FLOPs (dense -> MoE) | Dense | Pruned | MoE | MoE speedup |
> |---|---|---:|---:|---:|---:|---:|---:|
> | BAGEL single layer | generation MLP, 3e/1s/k1 | 32768 | 8.86 -> 4.43 TF | 74.185 ms | 36.336 ms | 38.983 ms | **1.903x** |
> | BAGEL single layer | generation MLP, 3e/1s/k1 | 65536 | 17.72 -> 8.86 TF | 150.980 ms | 73.524 ms | 77.830 ms | **1.940x** |
> | Qwen-Image single layer | MMDiT FFN, 3e/1s/k1 | 32768 | 4.95 -> 2.47 TF | 27.395 ms | 13.791 ms | 17.127 ms | **1.600x** |
> | Qwen-Image single layer | MMDiT FFN, 3e/1s/k1 | 65536 | 9.90 -> 4.95 TF | 54.206 ms | 28.852 ms | 35.648 ms | **1.521x** |
> | Ming single layer | MMDiT FFN, 3e/1s/k1 | 32768 | 1.24 -> 0.62 TF | 7.405 ms | 3.428 ms | 5.204 ms | **1.423x** |
> | Ming single layer | MMDiT FFN, 3e/1s/k1 | 65536 | 2.47 -> 1.24 TF | 14.875 ms | 7.706 ms | 11.580 ms | **1.285x** |
>
> ### Scope of MoE Adaptation
>
> The revision states that full training-aware MoE Adaptation is currently demonstrated on BAGEL / GenEval. The multi-model pruning, compression, and partition results are diagnostic evidence rather than a claim that the trained adaptation recipe has already transferred to every model.
>
> | Model | Training-free intervention | Component | GenEval | Takeaway |
> |---|---|---|---:|---|
> | BAGEL | 50% compression | all generation MLPs | 0.63 | full generation-MLP compression is relatively robust |
> | Qwen-Image | 50% compression | `txt_mlp` | **0.75** | promising text/context-side target |
> | Qwen-Image | 50% compression | `img_mlp` | 0.11 | image stream is more fragile |
> | Qwen-Image | 50% compression | all generator MLPs | 0.05 | uniform generator compression is unsafe |
> | Ming-Omni | 50% compression | `ff_context` | 0.19 | context stream is less damaging but still weak |
> | Ming-Omni | 50% compression | `ff` | 0.00 | image stream collapses |
> | Ming-Omni | 50% compression | all generator MLPs | 0.00 | uniform generator compression collapses |
>
> ### Reproducibility Details
>
> The revision consolidates implementation details that were previously scattered across the manuscript and scripts.
>
> | Reviewer point | Revision addition |
> |---|---|
> | Calibration size | Each task uses eight label-free calibration samples drawn from the same domain |
> | Routing granularity | Each converted MLP layer has one router with per-token top-\(k\) routed expert selection |
> | Top-\(k\) / shared experts | Exact total/shared/top-\(k\) settings are listed with the result tables |
> | Converted layers | Exact converted ranges are listed, e.g., BAGEL adaptation uses layers 1-27 |
> | Router initialization | Relaxed gate parameterization with zero-initialized shared-gate correction |
> | Training details | Adaptation data, optimizer, learning rates, token budget, auxiliary loss, and checkpoint selection are added |
>
> ### Baselines, Variance, and Calibration
>
> The revision cross-references existing baselines: dense fine-tuning, expert partitioning, expert-frozen tuning, full MoE adaptation, structured compression, and quantization. It also states when reported benchmark numbers are averaged across seeds and labels Geneval-derived pruning/partition calibration as task-calibrated or transductive when evaluation prompts are reused.
>
> The broader claim is therefore narrower and easier to verify: the paper diagnoses component-dependent sparsity across unified models, while full trained MoE Adaptation is reported for BAGEL / GenEval.

---

> > ### Author Response · Authors · 2026-06-03
> > **Follow-up to Reviewer bnKk**
> >
> > Thank you again for the careful and constructive assessment. We have updated the author response and planned revision around the main points you raised.
> >
> > In particular, we added direct latency / throughput / peak-memory measurements, clarified that full MoE Adaptation is currently scoped to BAGEL / GenEval, and consolidated key reproducibility details such as calibration size, routing granularity, top-\(k\), shared experts, and adaptation settings.
> >
> > We would greatly appreciate your feedback on whether these additions address your main concerns about efficiency evidence, scope, and reproducibility. We are happy to clarify any remaining point.
> >
> > Best,
> > Authors

---

### Author Response · Authors · 2026-03-26
**Gentle check-in on submission status**

Dear Action Editor,

We hope you are doing well.

This is a gentle note regarding our submission.

We noticed that the manuscript is currently in the "AE Assigned" stage and has not yet entered the "Under Review" phase. We completely understand that identifying suitable reviewers may take time.

This message is simply a brief check-in, and we truly appreciate your time and effort in handling the submission.

Thank you very much for your consideration.

---

### Author Response · Authors · 2026-05-22
**Summary of Revision**

We thank the reviewers for the constructive feedback. The revision strengthens the manuscript along five main axes.

## 1. Added Measured Efficiency Evidence

The revision adds direct runtime measurements instead of relying only on activated-parameter counts. We now report matched dense/pruned/MoE latency, throughput, peak memory, and target-component FLOP context. The end-to-end MoE measurements show positive speedups on BAGEL, Qwen-Image, and Ming, while the single-layer diagnostics explain how local sparse-compute savings are diluted by routing overhead and unconverted pipeline components.


## 2. Made the Generator Analysis Component-Aware

The revision separates generator subcomponents rather than treating all generation-side MLPs as one target. We added Qwen-Image image-stream versus text/context-stream results, Ming image-stream versus context-stream results, and Janus AR-language versus generation-head results. This makes the conclusion more precise: generator sparsity is module-dependent, so future adaptation targets should be selected by component.

## 3. Added AR-Generation Boundary Evidence

To address generality beyond diffusion-style generators, the revision adds a scoped Janus-Pro-7B analysis. These results extend the component-level diagnosis to autoregressive visual generation and show that AR-language MLPs and generation heads have distinct sparsity behavior, reinforcing the need for component-aware adaptation.

## 4. Clarified the Scope of MoE Adaptation

We sharpened the trained MoE Adaptation claim around the setting directly validated in the paper: BAGEL on GenEval. The broader multi-model results strengthen the central diagnosis by showing consistent component-dependent sparsity patterns across model families, and they provide concrete guidance for extending MoE Adaptation to additional architectures.

## 5. Improved Reproducibility and Presentation

The revision consolidates implementation details that were previously scattered across the manuscript and scripts, including calibration size, routing granularity, expert settings, converted layer ranges, router initialization, training details, and checkpoint selection. We also revised Figure 2 and its caption to explicitly define the understanding and generation top-neuron sets, making the overlap analysis easier to interpret.

Overall, the revision adds direct efficiency evidence, strengthens the evidence base for component-dependent sparsity, and makes the path from diagnosis to MoE Adaptation clearer and more reproducible.

---

> ### Author Response · Authors · 2026-06-08
> **Follow-up on Revised Manuscript**
>
> Dear Reviewers and Area Chairs,
>
> We would like to gently follow up on our revised manuscript. If there are any remaining concerns or points that would benefit from clarification, we would be happy to address them.
>
> Thank you again for your time.

---

### Author Response · Authors · 2026-05-29
**Follow-up on Author-Reviewer Discussion**

Dear Reviewers,

Thank you again for your time and for the constructive comments on our submission. We have responded to the points raised in the reviews and would greatly appreciate it if you could take a look when you have a chance.

Your feedback would be very helpful for us to better understand and address any remaining concerns. We would also be happy to provide further clarification if needed.

Thank you very much for your consideration.

Best,
Authors